# Cancer-specific PERK signaling drives invasion and metastasis through CREB3L1

Yu-Xiong Feng[1], Dexter X. Jin[1,2], Ethan S. Sokol[1,2], Ferenc Reinhardt[1], Daniel H. Miller[1,2] & Piyush B. Gupta[1,2,3,4]

PERK signaling is required for cancer invasion and there is interest in targeting this pathway for therapy. Unfortunately, chemical inhibitors of PERK's kinase activity cause on-target side effects that have precluded their further development. One strategy for resolving this difficulty would be to target downstream components of the pathway that specifically mediate PERK's pro-invasive and metastatic functions. Here we identify the transcription factor CREB3L1 as an essential mediator of PERK's pro-metastatic functions in breast cancer. CREB3L1 acts downstream of PERK, specifically in the mesenchymal subtype of triple-negative tumors, and its inhibition by genetic or pharmacological methods suppresses cancer cell invasion and metastasis. In patients with this tumor subtype, CREB3L1 expression is predictive of distant metastasis. These findings establish CREB3L1 as a key downstream mediator of PERK-driven metastasis and a druggable target for breast cancer therapy.

[1] Whitehead Institute for Biomedical Research, 455 Main Street, Cambridge, MA 02142, USA. [2] Department of Biology, Massachusetts Institute of Technology, Cambridge, MA 02139, USA. [3] David H. Koch Institute for Integrative Cancer Research, 500 Main Street, Cambridge, MA 02142, USA. [4] Harvard Stem Cell Institute, 7 Divinity Ave, Cambridge, MA 02138, USA. Yu-Xiong Feng and Dexter X. Jin contributed equally to this work. Correspondence and requests for materials should be addressed to P.B.G. (email: pgupta@wi.mit.edu)

The PERK kinase plays a critical role in tumor invasion and metastasis. PERK signaling—which is activated downstream of the unfolded protein response (UPR)[1, 2] and the integrated stress response[3]—enables cancer cells to survive the adverse conditions typically observed in tumor microenvironments[4–6]. Upon its activation, PERK phosphorylates eIF2a, which reduces global protein synthesis while selectively increasing the translation of ATF4[7]. In addition to enabling cell survival, PERK–ATF4 signaling triggers multiple steps in the metastatic cascade[8], including angiogenesis[9], migration[10], survival[11], and colonization at secondary organ sites[12]. PERK is also required for the metastatic dissemination of cancer cells that have undergone an epithelial-to-mesenchymal transition (EMT)[13].

Given its critical role in driving metastatic progression, PERK has been a focus of drug discovery programs for cancer, which have identified several small-molecule inhibitors of this kinase that reduce metastatic dissemination[13]. While these molecules reduce metastatic spread, they also cause rapid onset of pancreatic atrophy, precluding their further consideration for clinical development[14]. Since similar phenotypes are observed in PERK-knockout mice, PERK is likely to be essential for normal pancreatic function[15, 16]. This has raised concerns about whether this key pathway is a viable target for therapy. One way to circumvent this difficulty would be to target downstream factors that specifically mediate PERK's pro-metastatic functions, but do not contribute to the pathway's functions in noncancerous tissues. While this strategy is promising in principle, it is not currently known if the PERK pathway's pro-metastatic

functions can be genetically separated from its role in normal homeostasis.

In this study, we show that this is indeed the case, and identify a key transcription factor (CREB3L1) downstream in the pathway, that specifically functions to promote metastasis in cancer cells that have activated PERK. Since CREB3L1 must be proteolytically cleaved in order to become active, this provides a unique opportunity to target this factor with small-molecule inhibitors for therapy[17]. While CREB3L1's role in cancer is currently controversial—with some studies suggesting that this factor promotes metastasis, and others suggesting it suppresses metastasis[18, 19]—we resolve this discrepancy by showing that CREB3L1 specifically promotes metastasis in tumors that have activated both PERK signaling and the EMT program. Collectively, our findings establish CREB3L1 as a promising target downstream of the PERK pathway for therapeutic blockade in cancer.

## Results

**Cancer-specific PERK signaling correlates with metastasis.** Consistent with prior reports[13], we found that PERK is activated by phosphorylation in human breast cancers (Supplementary Fig. 1a). To identify factors downstream of PERK specifically upregulated in human breast cancers, we compared PERK pathway gene expression between a large cohort of breast cancers ($n = 1093$) and normal breast tissues ($n = 112$) (TCGA, breast cancer data set). Of the ~400 genes downstream of PERK[20] (Supplementary Data 1), only 23 showed at least a twofold

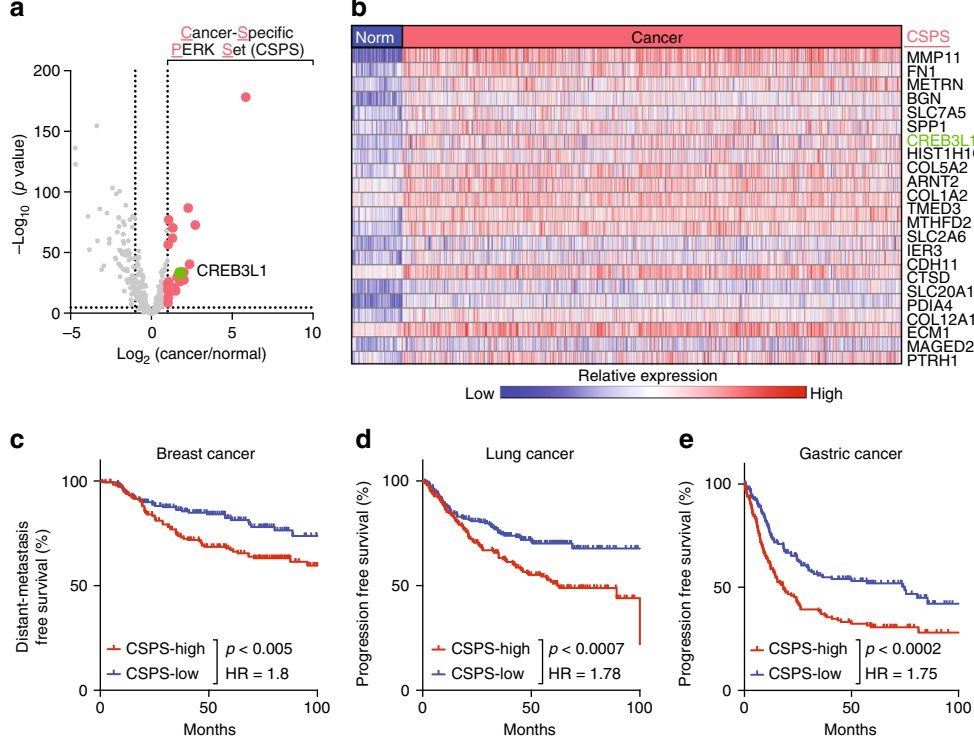

**Fig. 1** Cancer-specific PERK signaling correlates with progression and metastasis. **a** A volcano plot comparing the expression of 388 PERK pathway genes between breast cancer tissues and normal breast tissues. Plotted for each gene are the negative $\log_{10}$ of the $p$ value and the $\log_2$ of the fold change of gene expression of cancer samples relative to normal samples. A set of 23 genes that are twofold overexpressed in cancer and significant after Bonferroni correction were defined as the cancer-specific PERK set (CSPS; red dots). **b** A heat map showing the expression of the CSPS genes across 112 normal breast samples and 1093 breast tumors. **c–e** Average expression of CSPS genes was used to examine survival in breast, lung, and gastric tumor data sets. Patients were binned into CSPS high (top tertile) and CSPS low (bottom tertile), and survival curves were plotted. A cohort of **c** breast cancers was analyzed for metastasis-free survival (GSE17907, GSE9195, GSE20685, GSE16446, and GSE19615, $n = 182$ for each arm). A cohort of **d** lung cancers and **e** gastric cancers were analyzed for progression-free survival (lung cancer: GSE8894, GSE50081, GSE29013, and GSE31210, $n = 195$ for each arm; gastric cancer: GSE22377, GSE15459, and GSE62254, $n = 167$ for each arm). The indicated $p$-values were calculated using the log-rank test

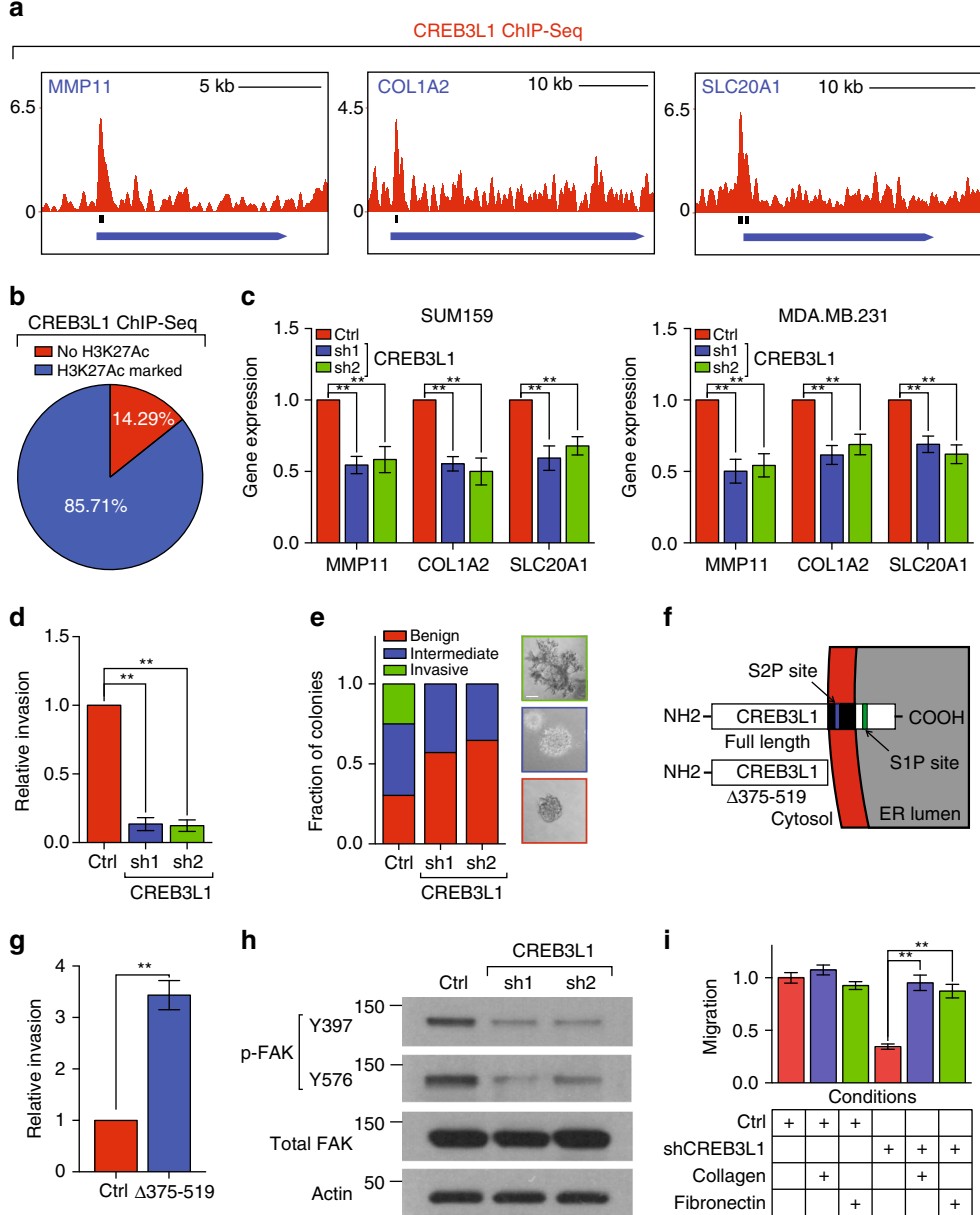

**Fig. 2** CREB3L1 upregulates cancer-specific PERK genes and is required for invasion. **a** Distribution of CREB3L1 ChIP-seq fold-enrichment over input control, along with called-peaks, at the locus of CSPS genes. Shown were the CREB3L1 peaks for three CSPS genes: MMP11, COL1A2, and SLC20A1. **b** A pie chart showing the fraction of CREB3L1 ChIP-Seq peaks from the CSPS genes that were co-localized with acetylated H3K27. **c** qPCR showing the relative expression of the CSPS genes shown in **a** in SUM159 and MDA.MB.231 cells transduced with shRNAs targeting luciferase or CREB3L1, $n = 4$. **d** Quantification of the cell invasion of SUM159 cells transduced with shRNAs targeting luciferase or CREB3L1 in a basement membrane-coated transwell assay, $n = 3$. **e** Left panel: quantification of colony types formed by SUM159 cells transduced with shRNAs targeting luciferase or CREB3L1 in a 7-day 3D Matrigel invasion assay, $n = 50$; right panel: representative images of benign, intermediate, and invasive colonies (scale bar: 80 μm). **f** Schematic of full-length CREB3L1 and CREB3L1$^{\Delta375-519}$. **g** Quantification of cell invasion of HMLE cells transduced with control plasmid or a constitutive active form of CREB3L1 (CREB3L1$^{\Delta375-519}$) in a basement membrane-coated transwell assay, $n = 3$. **h** Western blotting of phospho-FAK (pY397 and pY576), total FAK, and Actin in SUM159 cells transduced with shRNAs targeting luciferase or CREB3L1. **i** Quantification of cell migration of SUM159 cells transduced with shRNAs targeting luciferase or CREB3L1 treated with the indicated conditions, $n = 8$. Data are represented as mean ± SEM. * indicates $p < 0.05$; ** indicates $p < 0.01$ (Student's $t$-test)

increase in expression in cancers relative to normal tissues (Fig. 1a, b; CSPS, cancer-specific PERK set). Expression of these genes depended on PERK activity, since inhibition of PERK with a small-molecule inhibitor (GSK2656157[14]) led to a significant decrease in the expression of 18 out of 23 CSPS genes (Supplementary Fig. 1b). Breast cancers that highly expressed these cancer-specific PERK genes were significantly more likely to develop distant metastases over 10 years (HR = 1.8, $p < 0.005$,

log-rank test; Fig. 1c). Expression of these genes was also predictive of tumor progression in lung and gastric cancers (Fig. 1d, e).

**CREB3L1 upregulates CSPS genes and is required for invasion.** CREB3L1 was the only transcription factor among the cancer-specific PERK genes we identified (Fig. 1a, b). Analysis of ChIP-seq data indicated that CREB3L1 was enriched near CSPS gene loci (14/23 genes, $p < 0.01$, binomial test; Fig. 2a and

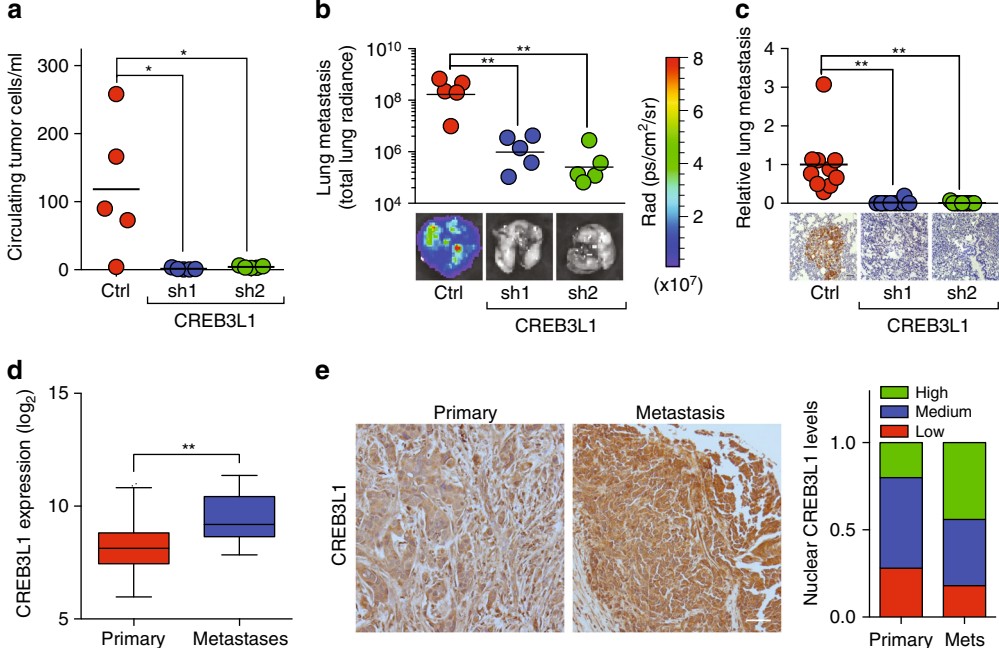

**Fig. 3** CREB3L1 is required for metastasis and is upregulated in the metastases of breast cancer patients. **a** A quantification of circulating tumor cells (CTCs) per ml of blood from NOD/SCID mice 7 weeks after being transplanted with MDA.MB.231-Luc-LM2 cells transduced with shRNAs targeting LacZ or CREB3L1, $n = 5$. **b** Lung metastases of the animals in **a** were quantified (top) by radiance of the whole lung in each animal, $n = 5$. Total radiance was also imaged (bottom). **c** Lung metastases of the animals in **a** was gauged by immunohistochemistry staining for GFP-positive cancer cells in lung sections. GFP area per field was quantified (top) and imaged (bottom), $n = 10,10,9$ for Ctrl, sh1, and sh2, respectively (scale bar: 40 μm). **d** Expression of CREB3L1 in 529 primary breast cancer tissues and 45 breast cancer metastases (GSE20565, GSE20685, GSE7904, and GSE3744) shown as a box-and-whiskers plot. **e** Immunohistochemical staining for CREB3L1 in a tissue microarray containing primary breast cancer tissues (left, $n = 50$) and breast cancer metastases in lymph nodes (middle, $n = 50$). Nuclear CREB3L1 levels were quantified (right). Scale bar: 40 μm. Data are represented as mean ± SEM or the mean alone. * indicates $p < 0.05$; ** indicates $p < 0.01$ (Student's $t$-test)

Supplementary Fig. 2a); CREB3L1 was localized to active regulatory regions marked by H3K27-acetylated histones (86% of bound sites; Fig. 2b)[21]. To functionally assess CREB3L1's role in regulating these genes, we used shRNAs to stably inhibit CREB3L1 expression in two invasive breast cancer cell lines, SUM159 and MDA.MB.231 (Supplementary Fig. 2b). Inhibiting CREB3L1 reduced the expression of 10 of these 14 CSPS genes (Fig. 2c and Supplementary Fig. 2c). Taken together, these data indicated that CREB3L1 directly promotes the transcription of CSPS genes.

We next used three approaches to assess if CREB3L1 was required for cancer cell invasion. First, we compared CREB3L1 expression between non-invasive and invasive breast cancer lines[22, 23], which revealed that this factor's expression was increased over 50-fold in the invasive lines (Supplementary Fig. 3a). Second, we inhibited CREB3L1 expression to test if it was required for the invasiveness of invasive cancer lines. In both the SUM159 and MDA.MB.231 lines, CREB3L1 inhibition caused a sixfold reduction in cellular invasion through basement membrane-coated transwell chambers (Fig. 2d and Supplementary Fig. 3b). In contrast, knockdown of CREB3L1 does not affect cell growth of SUM159 and MDA.MB.231 cells (Supplementary Fig. 3c, d). Lastly, we tested the requirement of CREB3L1 for invasion in 3D. When seeded as single cells into 3D cultures in basement membrane, SUM159 and MDA.MB.231 breast cancer lines form clonal structures that can be classified as benign (non-invasive), intermediate (partially invasive), or invasive. Knockdown of CREB3L1 prevented the formation of invasive spheroids, while increasing the formation of non-invasive spheroids (Fig. 2e and Supplementary Fig. 3e).

CREB3L1 is localized to ER membranes and can migrate to the nucleus to activate transcription. Its nuclear migration is triggered

via cleavage by the site 1 (S1P) and site 2 proteases (S2P), which produces an activated, truncated form of CREB3L1[24]; the truncated CREB3L1 is then free to migrate into the nucleus and activate transcription (Fig. 2f). Both the full length and truncated form of CREB3L1 were upregulated in invasive cancer cells when compared to non-invasive cells (Supplementary Fig. 3f). We found that the aforementioned decrease in invasive spheroids following knockdown of CREB3L1 is rescued by overexpression of a constitutively active form of CREB3L1 (CREB3L1$^{\Delta375-519}$; Supplementary Fig. 3g, h). Next, we examined if CREB3L1 was sufficient to promote invasion in non-invasive cells. Overexpression of CREB3L1$^{\Delta375-519}$ resulted in a fourfold increase in cellular invasion in a non-invasive HMLE cell line (Fig. 2g), indicating that CREB3L1's expression was sufficient to promote invasion.

**CREB3L1-induced ECM deposition activates FAK.** To further explore the mechanism through which CREB3L1 promotes invasion, we performed gene ontology analysis of CREB3L1-regulated CSPS genes. The analysis suggested that CREB3L1 might regulate ECM production and remodeling (Supplementary Fig. 4a; $p < 10^{-5}$, hypergeometric test), key processes associated with cancer cell invasion. Consistent with this, CREB3L1 inhibition strongly reduced activation of FAK—a kinase regulated by cell–ECM interactions and known to be important for cellular migration (Fig. 2h)[25, 26]. To assess if CREB3L1 was important for migration, we seeded CREB3L1-inhibited or control cells into transwell chambers. We found that CREB3L1 inhibition caused a fivefold reduction in migratory potential (Supplementary Fig. 4b). Since CREB3L1 regulates the expression of ECM genes, e.g., *COL1A2* and *FN1* (Fig. 2c and Supplementary Fig. 2c), we

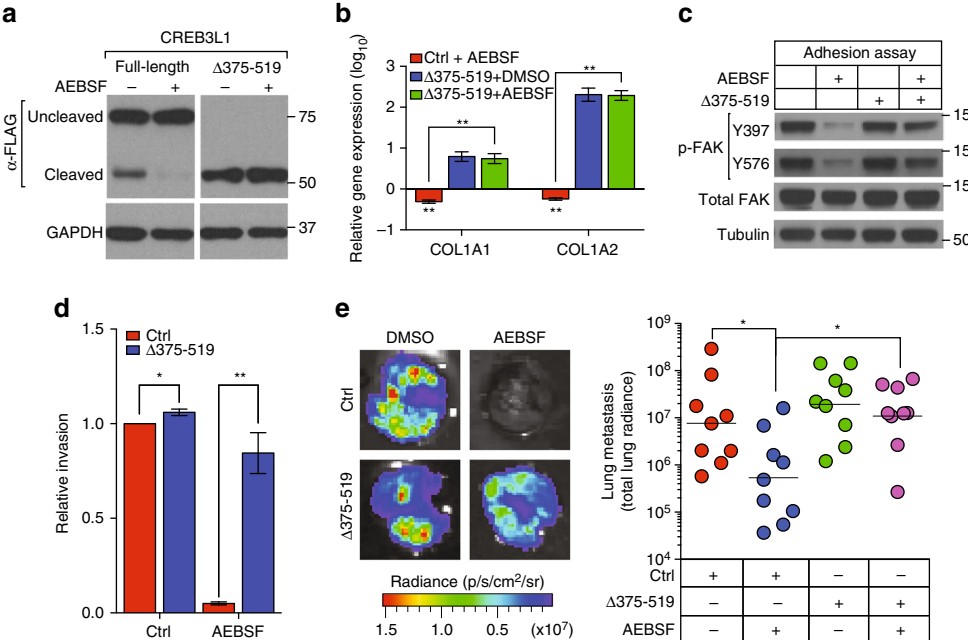

**Fig. 4** Chemical inhibition of CREB3L1 blocks metastasis. **a** A western blot of CREB3L1 expression in full-length CREB3L1 or CREB3L1$^{\Delta375-519}$ transduced SUM159 cells that were treated with solvent control or a S1P inhibitor, AEBSF. **b** SUM159 cells transduced with control or CREB3L1$^{\Delta375-519}$ were treated with solvent control or AEBSF. qPCR analyses were performed to quantify the expression of ECM genes: COL1A1 and COL1A2. The gene expression was normalized to the solvent-treated control transduction samples, $n = 3$. **c** Western blot of phosphorylated FAK (pY397 and pY576), total FAK, and Tubulin in cells of **b**. **d** Quantification of cell invasion of cells from **b** in a basement membrane-coated transwell assay, $n = 3$. **e** Two weeks after orthotopical transplantation with control or CREB3L1$^{\Delta375-519}$ transduced MDA.MB.231-Luc-LM2 cells, NOD/SCID mice were treated with solvent control or AEBSF for another 4 weeks. Lung metastases of the animals were gauged by radiance from the whole lung in each animal, $n = 9$. Data are represented as mean ± SEM or geometric mean alone. * indicates $p < 0.05$; ** indicates $p < 0.01$ (Student's $t$-test)

hypothesized that its inhibition could be reducing FAK activity by decreasing ECM production. Consistent with this, addition of type I collagen (encoded by COL1A1 and COL1A2) rescued FAK activity in the CREB3L1-inhibited cells (Supplementary Fig. 4c). If CREB3L1's pro-invasive effects were arising through ECM production, it should be possible to rescue the migration of CREB3L1-inhibited cells by supplementing them with exogenous pro-invasive ECM. In fact, cells inhibited for CREB3L1 expression were able to better heal wounds when treated with exogenous type I collagen or fibronectin (Fig. 2i). Collectively, these results suggested that CREB3L1 may promote migration and invasion by inducing ECM production.

**CREB3L1 is required for breast cancer metastasis**. We next assessed CREB3L1's role in vivo using the MDA.MB.231-LM2 orthotopic transplantation model of human breast cancer, which forms tumors that invade the vasculature and metastasize to the lungs. While CREB3L1 inhibition does not affect primary tumor formation or growth, as indicated by similar tumor weights and Ki67-positive proliferative indices (Supplementary Fig. 5), there was a >80-fold reduction in the number of circulating tumor cells (CTCs) (Fig. 3a). Moreover, CREB3L1 inhibition resulted in a >400-fold reduction in lung metastases, as measured both by luminescence emitted by metastatic cells in whole lungs, as well as by immunohistochemistry staining for GFP-positive cancer cells in lung sections (Fig. 3b, c). These findings indicated that CREB3L1 was required for cancer cells to enter into the circulation and form metastases.

To assess the clinical relevance of these findings, we examined CREB3L1 expression in primary breast cancers and metastatic breast cancers from patients. In comparison with primary breast cancer tissues, CREB3L1 gene expression was significantly

upregulated in the metastatic growths of breast cancers that had disseminated to distant organ sites ($p < 0.01$, Student's $t$-test; Fig. 3d, GSE20565, GSE20685, GSE7904, and GSE3744). Consistent with these findings, nuclear and total CREB3L1 protein were expressed at significantly higher levels in the lymph node metastases of breast cancers, relative to primary breast tumors present on the same tissue microarray (Fig. 3e).

These findings indicated that CREB3L1 is required for metastatic dissemination in animal models of breast cancer, and is upregulated in the breast cancers of patients as they progress toward metastatic disease.

**Chemical inhibition of CREB3L1 blocks metastasis**. Although most transcription factors are currently not "druggable", CREB3L1's unique mechanism of activation provides an opportunity to target its activity using protease inhibitors. A chemical inhibitor of proteases, AEBSF, has been used previously to inhibit the protease (S1P) that cleaves CREB3L1 to its active form[24, 27]. In SUM159 cells, AEBSF treatment blocked cleavage of FLAG-CREB3L1$^{full-length}$ protein to its active form, but had no effect on the levels of FLAG-CREB3L1$^{\Delta375-519}$ protein (Fig. 4a). Consistent with this, AEBSF treatment decreased the expression of CREB3L1 target genes COL1A1[28] and COL1A2 (Fig. 4b). However, AEBSF treatment did not decrease the expression of these genes in cells expressing the constitutively active FLAG-CREB3L1$^{\Delta375-519}$ (Fig. 4b). While this finding was expected, because this truncated protein does not need to be cleaved to become active, it nonetheless indicated that AEBSF's effects on COL1A1 and COL1A2 were mediated by CREB3L1. Inhibition of CREB3L1 activation by another S1P inhibitor, PF429242, also decreased the expression of COL1A1 and COL1A2, which could be rescued by expression of FLAG-

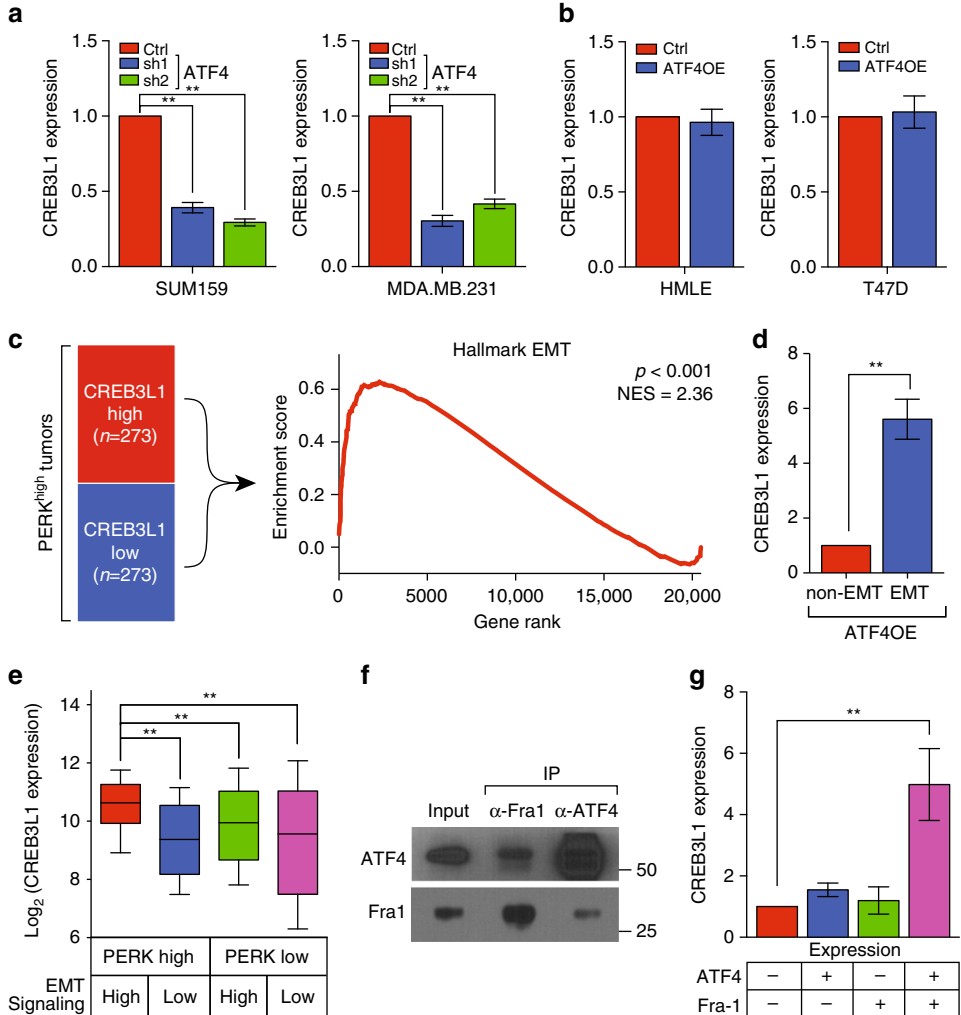

**Fig. 5** PERK signaling requires an EMT to upregulate CREB3L1. **a** qPCR showing the relative expression of CREB3L1 in SUM159 and MDA.MB.231 cells transduced with shRNAs targeting luciferase or ATF4, $n = 3$. Data are represented as mean ± SEM. * indicates $p < 0.05$; ** indicates $p < 0.01$ (Student's $t$-test). **b** qPCR showing the relative expression of CREB3L1 in HMLE and T47D cells transduced with a control plasmid or ATF4-expressing plasmid (ATF4OE), $n = 3$. Data are represented as mean ± SEM. * indicates $p < 0.05$; ** indicates $p < 0.01$ (Student's $t$-test). **c** A set of 546 breast cancers (TCGA) expressing high levels of PERK pathway genes (PERK[high] tumors, see "Methods" section for details) were stratified into CREB3L1-high and CREB3L1-low cohorts, according to the expression of CREB3L1. Gene set enrichment analysis using the hallmark gene sets was performed to compare the CREB3L1-high and CREB3L1-low tumors (left). The top hallmark gene set was an EMT set (right). **d** qPCR showing the relative expression of CREB3L1 in HMLE cells overexpressing ATF4 with or without induction of an EMT, $n = 3$. Data are represented as mean ± SEM. * indicates $p < 0.05$; ** indicates $p < 0.01$ (Student's $t$-test). **e** Expression of CREB3L1 in breast cancers (TCGA) that have differential levels of PERK signaling and EMT signaling as indicated. Data are represented as a box-and-whiskers plot, $n = 122$. * indicates $p < 0.05$; ** indicates $p < 0.01$ (Mann–Whitney $U$-test). **f** HMLE cells were transduced with ATF4 and Fra-1 expressing constructs. Western blot for ATF4 or Fra-1 after immunoprecipitation of either ATF4 or Fra-1, which shows that these factors co-immunoprecipitate each other. **g** qPCR showing the relative expression of CREB3L1 in HMLE cells overexpressing control, ATF4, Fra-1, or a combination of ATF4 and Fra-1, $n = 3$. Data are represented as mean ± SEM. * indicates $p < 0.05$; ** indicates $p < 0.01$ (Student's $t$-test)

CREB3L1$^{\Delta 375-519}$ (Supplementary Fig. 6a). In addition, AEBSF treatment strongly reduced FAK phosphorylation, which could be restored by expression of FLAG-CREB3L1$^{\Delta 375-519}$ (Fig. 4c). Furthermore, inhibition of CREB3L1 activation by AEBSF or PF429242 significantly negated the invasiveness of the SUM159 cells, an effect that can be rescued by overexpression of the truncated FLAG-CREB3L1$^{\Delta 375-519}$ (Fig. 4d and Supplementary Fig. 6b–d). In contrast, overexpression of the constitutively active form of two other targets of AEBSF, ATF6 and SREBP-1, was not able to rescue the decrease of cell invasion caused by AEBSF (Supplementary Fig. 6e, f). These results showed that AEBSF inhibits invasion through targeting CREB3L1.

We next assessed if treatment with AEBSF could inhibit metastasis in vivo in the orthotopic MDA.MB.231-LM2 model of human breast cancer. While AEBSF did not affect primary tumor growth (Supplementary Fig. 7a, b), it caused a 15-fold reduction in lung metastases, as measured by luminescence emitted from whole lungs (Fig. 4e). However, AEBSF was not able to inhibit the metastasis of LM2 tumors that overexpressed FLAG-CREB3L1$^{\Delta 375-519}$, indicating that its effects were mediated through CREB3L1. These observations suggest that inhibiting S1P activity to block CREB3L1 activation could be an effective therapeutic strategy for preventing metastasis.

**PERK signaling requires an EMT to upregulate CREB3L1.** Lastly, we investigated how this factor is regulated in invasive cancer cells. Consistent with the finding that CREB3L1 is downstream of PERK signaling (Supplementary Fig. 8), inhibition of ATF4 expression decreased CREB3L1 transcription in MDA.

MB.231 and SUM159 lines (Fig. 5a). Surprisingly, overexpression of ATF4 in non-invasive lines (HMLE and T47D) is not sufficient to activate transcription of CREB3L1, suggesting that PERK signaling requires additional pathways to induce CREB3L1 (Fig. 5b). To explore these additional pathways, we segregated breast cancers that have active PERK signaling into two categories according to the expression level of CREB3L1 (CREB3L1-high and CREB3L1-low, respectively), and employed gene set enrichment analysis (GSEA) to compare these two cohorts (Fig. 5c). We found that the EMT hallmark set was the most enriched pathway in tumors that had high CREB3L1 expression relative to those with low CREB3L1 expression (Fig. 5c). This implicated EMT signaling as a potential regulator of CREB3L1 expression in the context of PERK activation. To functionally assess this, we assayed CREB3L1 expression with or without EMT induction in cells overexpressing ATF4. We found that HMLE cells induced into EMT had a greater than fivefold higher CREB3L1 expression than those that were not (Fig. 5d). Consistently, expression of CREB3L1 was highest in breast tumors that have both active PERK signaling and EMT signaling (Fig. 5e). Furthermore, expression of CREB3L1 is predictive of distant metastasis-free survival in patients with triple-negative breast cancer of the mesenchymal subtype, which is usually enriched for pathways associated with EMT (Supplementary Fig. 9)[29].

We next further dissected which component of EMT signaling cooperates with PERK signaling to induce CREB3L1. Using the proteomic interaction database, Biogrid, we found that ATF4 might interact with Fra-1, a potent EMT transcription factor[30, 31]. By performing co-immunoprecipitation, we validated that ATF4 interacts with Fra-1 in HMLE cells (Fig. 5f). Double overexpression of ATF4 and Fra-1 strongly induces the expression of CREB3L1 in non-EMT HMLE cells, while single overexpression of either gene is not sufficient to induce CREB3L1 (Fig. 5g and Supplementary Fig. 10). These results suggested that PERK signaling requires EMT to upregulate CREB3L1 through an ATF4–Fra-1 interaction.

## Discussion

The present findings establish CREB3L1 as a key downstream mediator of PERK's pro-metastatic function in cancers. CREB3L1's unique mode of activation makes it amenable—unlike the vast majority of transcription factors—to inhibition by small molecules. Thus, our findings identify a viable new avenue for therapeutically targeting the PERK pathway in cancer.

The phenotypes observed in prior studies with transgenic mice provide strong support for the notion that targeting CREB3L1 would circumvent the toxicities associated with PERK inhibitors. While CREB3L1-knockout mice only have bone defects[28], PERK-knockout mice exhibit not only these bone defects, but also reduced birth rates and develop diabetes[15, 16, 32, 33]. Additionally, mice lacking ATF4—which is upstream of CREB3L1 but downstream of PERK—also have bone defects, lower birth rates, and develop anemia[34–37]. These additional deficiencies strongly suggest that both PERK and ATF4 have CREB3L1-independent functions. In fact, in addition to activating CREB3L1, PERK also activates NRF2—a transcriptional regulator of the cellular antioxidant response[38]—whereas ATF4 also activates the transcription of CHOP and BiP[39]. These observations are in consonance with the general principle that upstream components of signaling pathways typically have more pleiotropic effects when compared to downstream components.

While our results provide a proof-of-concept for inhibiting PERK-CREB3L1 signaling in vivo, the use of site 1 protease (S1P) inhibitors may raise concerns regarding drug specificity. Three additional ER-bound transcription factors, ATF6 and SREBP-1/2, are also affected by agents that inhibit S1P/S2P-mediated proteolysis[40]. However, our data indicate that these factors are not required for cell invasion, precluding their role in mediating the anti-invasive effects of AEBSF in our experiments. Moreover, Nelfinavir, a S2P inhibitor used in treating HIV patients, has been well tolerated in clinical trials[41], indicating that inhibition of ATF6 and SREBP-1 is not likely to be problematic in patients.

Defects in ECM remodeling are hallmarks of human diseases—including cancer and fibrosis[42, 43]—and are frequently associated with increased UPR signaling[2, 44]. Our findings provide the first molecular explanation for how PERK might regulate ECM remodeling, namely through activation of CREBL1. Consistent with this model, S1P inhibition—which would inhibit CREB3L1—has also been shown to significantly decrease expression of ECM genes, including COL1A1, COL1A2, and FN1[27]. This model also explains the bone defects observed in CREB3L1, ATF4, and PERK-knockout mice. The primary function of osteoblasts is to produce ECM, which spontaneously mineralizes to form bone[45]. Our results would suggest that these bone defects arise because osteoblasts in CREB3L1, ATF4, and PERK-knockout mice produce insufficient ECM[36, 37]. In fact, a prior study has shown that expression of COL1A1 and COL1A2 is significantly reduced in CREB3L1-knockout mice[28]—consistent with our finding that CREB3L1 regulates these collagens in invasive cancer cells.

Our observations indicate that the signaling downstream of PERK activation is highly dependent on the differentiation state of cancer cells. Thus, while PERK induces CREB3L1 in cancer cells that have undergone an EMT, it fails to do so in cells that have not undergone an EMT. Consistent with this, although PERK is required for invasion in cancer cells that have undergone an EMT, its activation is insufficient to drive invasion in a non-EMT context[13]. This suggests that CREB3L1 specifically promotes metastasis in tumors that have activated both PERK signaling and the EMT program. This observation may provide an explanation for the conflicting reports in the literature regarding CREB3L1's role in metastasis[18, 19]. More generally, our findings suggest that the ability of stress signaling to promote metastasis will depend on the differentiation state of tumors.

Our observations provide a new avenue for treating the mesenchymal subtype of triple-negative breast cancers (TNBC). As a class, TNBCs afford a relatively poor prognosis, and are therefore the focus of molecular subtyping and drug development efforts[29, 46]. We have found that CREB3L1 expression strongly correlates with the metastatic potential of this subtype of TNBCs. Since the mechanism through which CREB3L1 is activated can be effectively targeted by small-molecule inhibitors, our findings suggest a promising new therapeutic strategy for this clinically important subtype of breast cancer.

## Methods

**Human cell lines and culture conditions**. MCF7, T47D, BT474, BT549, ZR-75-30, Hs578T, MDA-MB-157, and MDA.MB.231 were obtained from the American Type Culture Collection (ATCC), and were cultured in DMEM + 10% FBS. SUM159 cells were obtained from Asterand, and were cultured in F12 + 5% FBS, insulin (10 µg/ml), and hydrocortisone (0.5 µg/ml). MDA.MB.231-luc-LM2 (LM2) cells were a kind gift from Dr Joan Massagué and cultured in DMEM + 10% FBS. HMLE cells expressing the coding sequence of Twist1 fused to the mutated estrogen receptor (HMLE-TwER) were obtained from Dr Robert A. Weinberg's lab, and maintained in a 1:1 mixture of DMEM + 10% FBS, insulin (10 µg/ml), hydrocortisone (0.5 µg/ml), EGF (10 ng/ml), and MEGM. To induce an EMT, the HMLE cells were treated with 25 nM of 4-hydroxy-tamoxifen (4-OHT) for a period of 12 days.

**Lentiviral production and infection**. Lentiviral particles were produced by co-transfection with 0.25 µg pCMV-VSV-G, 0.75 µg pCMV-dR8.2-dvpr, and 1 µg pLKO-shRNAs into 5 × 10⁵ 293T cells using 6 µl Fugene 6. Cell culture media of 293T cells were harvested 24, 48, and 72 h post transfection. After harvesting, lentiviral transduction was performed in the presence of protamine sulfate by spin

infecting for 1 h, then incubating overnight. Following transduction, cells were selected using puromycin or blasticidin.

**Plasmids**. pCW-ATF4 was generated by digesting pCW57.1 (Addgene plasmid #41393) with NheI and AgeI. ATF4-3xFlag-V5 was amplified with NheI and AgeI overhangs and digested to insert into cut pCW57.1. pCW-CREB3L1$^{\Delta 375-519}$ was generated by cloning CREB3L1 without the amino acids between position 375 and the C-terminus with NheI and AgeI overhangs, digested, and inserted into pCW57.1 cut with NheI and AgeI. shATF4 and shCREB3L1 plasmids were from the Broad Institute TRC platform. pLX304-FOSL1-3xFlag-V5 was generated by digesting pLX304 (Addgene plasmid #25890) with BamHI and NheI. FOSL1-3xFlag-V5 was amplified with BamHI and NheI overhangs and digested to insert into cut pLX304. Cleaved N-terminus of ATF6 (1-373) and cleaved N-terminus of SREBP-1-c (1-490) were from Addgene (Ron Prywes lab, Addgene plasmid #27173, and Timothy Osborne lab, Addgene plasmid #26802). The pCW57.1 plasmid was a gift from David Root (Addgene plasmid #41393) and pLKO.1-blast plasmid was a gift from Keith Mostov (Addgene plasmid #26655).

**Western blot and co-immunoprecipitation**. Western blotting was performed according to a previous study with the following modifications[13]. Cultured cells were washed twice with PBS and lysed in radioimmunoprecipitation assay buffer for 15 min on ice. Cell lysates were clarified by centrifugation at 12,000×$g$ for 10 min, and protein concentration was determined by the BCA Reagent. Lysates were separated on NuPAGE 4–12% Bis-Tris gel electrophoresis, proteins were then transferred to nitrocellulose membrane and immunoblotted with indicated antibodies. All immunoblots were visualized by enhanced chemiluminescence. Antibodies used for immunoblotting were as follows: ATF4 (Cell Signaling, 11815, 1:1000), Flag (Sigma, M2 clone, 1:10,000), GAPDH (Cell Signaling, 3683, 1:2000), p-FAK (Cell Signaling, 8556 and 3281, 1:1000), Total FAK (Cell Signaling, 13009, 1:2000), and Fra-1 (Cell Signaling, 5281, 1:1000). In collagen rescue experiments, culture plates were coated with type I collagen (EMD, CC050) for 1 h at room temperature. Subsequently, $10^5$ cells were seeded onto the collagen and grown for 24 h before protein harvest. In co-immunoprecipitation experiments, $5 \times 10^7$ cells were seeded 24 h prior to harvesting. Whole cell lysate was collected and incubated with 10 µl anti-Flag antibodies and 50 µl Dynabeads Protein G, or 10 µl anti-Fra-1 antibodies and 50 µl Dynabeads Protein A, respectively, for 24 h at 4 °C. Immunoprecipitants were eluted and analyzed by western blot. Raw data of all western blots were included in Supplementary Fig. 11.

**In vitro transwell assay**. To assay migratory and invasive potential, we used a 24-well 8-µm pore transwell plate coated without or with Matrigel, respectively (Corning Inc.). Briefly, we seeded $5 \times 10^4$ cells in the upper chamber, and incubated the cells for 8 h before removing cells on the upper surface of the chamber to count the number of cells that migrated onto the lower surface of the chamber. We use crystal violet staining (Sigma-Aldrich) to label cells, and imaged five random fields per well with a microscope at ×10. Images were then quantified using ImageJ.

**In vitro wound healing migration assay**. Total of $5 \times 10^5$ cells were seeded in six-well plate 12 h prior to wound cutting. A 200 µl pipette tip was used to make scratches on the single cell layer. Images of cells were taken immediately or 8 h after wound cutting. All images were analyzed using ImageJ.

**3D invasion assay**. Basement membrane remodeling was assessed by seeding cells in growth factor reduced Matrigel (BD Biosciences). Matrigel was thawed at 4 °C. Then 100 µl of Matrigel was added to each chamber and allowed to solidify at 37 °C with 5% $CO_2$ in a humidified incubator for 30 min. A total of 500 single cells in a 200 µl suspension was seeded onto the initial layer of Matrigel and allowed to settle for 20 min. An aliquot of 200 µl of media supplemented with 10% Matrigel was gently layered on top. An aliquot of 200 µl of media was replaced every 4 days. Cultures were fixed and imaged 7 days after growth. Spheres were binned into three categories based on invasiveness: benign, intermediate, and invasive. Benign spheres were defined as structures with no visible cellular protrusions, intermediate spheres were defined as structures with protrusions extending from a central colony, and invasive structures were defined as fully scattered colonies.

**Spontaneous metastasis assay**. NOD/SCID mice were purchased from Jackson Labs. All mouse procedures were approved by the Animal Care and Use Committees of the Massachusetts Institute of Technology. For the spontaneous metastasis assay, $1 \times 10^6$ MDA.MD.231-LM2 cells were suspended in 50 µl 1:1 mix of Matrigel and DMEM, and injected into a mammary pad of each mouse. Animals were randomized by weight before treatment. After 7 weeks, mice were weighed. Tumors and lungs were harvested 7 weeks after injection. To measure metastases, freshly collected lungs were soaked in D-Luciferin (150 µg/ml; PerkinElmer) for 15 min and imaged with the IVIS system (PerkinElmer) at non-saturating exposures. Radiance was quantified with LivingImage v4.4 software. Tumors were weighed after collection.

**Drug treatment**. In the chemical treatment experiment, 2 weeks after tumor cells implantation, 200 µl pure PBS or PBS with 1 mg of AEBSF was administrated daily through i.p. injection into the animals for a period of 4 weeks.

**Immunohistochemistry and human tissue microarray**. All immunohistochemistry was performed at the Koch Institute Histology Core using the Thermo Scientific IHC Autostainer 360 (Thermo). Tumors obtained from the spontaneous metastasis assay were fixed in 10% neutral-buffered formalin, and then embedded in paraffin. Paraffin-embedded tumors were sectioned at 5 µm for histological analyses. Immunohistochemistry for Ki67 and GFP was performed. Stained sections were imaged and quantified. Briefly, two fields were scored per animal, resulting in a total of 10 fields per group. On each cross-section, ~5% of the total area was scored for GFP positivity using ImageJ. Scores were normalized to the mean of the control group to calculate relative lung metastasis. Tissue microarrays were ordered from Biomax (BR10010d). Immunohistochemistry for CREB3L1 and phosphorylated PERK (pPERK) was performed. CREB3L1 and pPERK staining was independently quantified in a blinded manner. Antibodies used in IHC were: GFP (Cell Signaling, 2956, 1:100), CREB3L1 (R&D systems, AF4080, 1:100), Ki67 (Cell Signaling, 9449, 1:250), and pPERK (PIERCE CHEMICAL MS, PA537773, 1:100).

**Quantification of circulating tumor cells**. A total of $1 \times 10^6$ luciferase-labeled MDA.MB.231-LM2 cells was inoculated into the fat pad of NOD/SCID mice. Seven weeks post tumor implantation, 400 µl of peripheral blood samples was collected. Blood samples were centrifuged at 1000×$g$ to precipitate the cell fractions. An aliquot of 1 ml of red blood cell lysis buffer (Stemcell Technology) was applied to resuspend the pellet on ice for 5 min. The samples were then centrifuged again at 2000 rpm. An aliquot of 200 µl PBS containing luciferin was used to resuspend the pellet, and luciferase activity of all samples were immediately measured using the IVIS imaging system. To quantify the number of luciferase-positive cells, different numbers of MDA.MB.231-LM2 cells, 1000, 333, 111, 37, and 12, were mixed with 400 µl of blood samples from tumor-free NOD/SCID mouse, respectively, and followed by the same purification and measurement processes mentioned above. A linear relationship between luciferase activity and cell number was then generated by linear regression, and the number of luciferase-positive circulating tumor cells from tumor-bearing mice were then calculated.

**GSEA and gene ontology analysis**. We defined the PERK gene set as the top 400 genes downregulated in an ATF4 knockout MEFs relative to wild-type MEFs following thapsigargin treatment (GSE35681). Gene symbols were converted to human genes that overlapped with those in the TCGA data set, which left 388 remaining symbols. Gene ontology enrichment analysis for cellular components was performed using the molecular signature database. Gene set enrichment analysis was performed using the GSEA software developed by the Broad Institute. PERK$^{high}$ patients were defined as those whose scores for the PERK gene set ranked in the top half. Subsequently, these PERK$^{high}$ patients were ranked based on CREB3L1 expression. For each gene, expression from the top half (CREB3L1$^{high}$) of PERK$^{high}$ tumors were compared to those from the bottom half (CREB3L1$^{low}$) of PERK$^{high}$ tumors. Gene set enrichment analysis was performed using hallmark gene sets provided by the molecular signature database on the ranked list of genes produced by comparing CREB3L1$^{high}$ to CREB3L1$^{low}$.

**Chromatin immunoprecipitation**. Chromatin immunoprecipitation was previously performed by the ENCODE Project using a CREB3L1 antibody (Sigma-Aldrich, HPA024069) as described[21]. Library was downloaded from ascension ENCLB300HOB. Reads were aligned to the human genome build GRCh38 with up to one mismatch. Chip-seq peaks were identified using MACS2 with a $q$ value threshold of 0.01. $q$ value was calculated using the Benjamini-Hochberg procedure. Each peak was annotated with the closest associated gene using HOMER. CSPS bound genes were considered overlapping with H3K27ac if they directly overlapped an H3K27ac peak. Distribution, as a fold change of CREB3L1 signal relative to input signal, and peaks were displayed using UCSC genome browser for hg38.

**Primary tumor and metastasis gene expression analyses**. Normalized RNA-seq data were obtained from https://gdac.broadinstitute.org/. Data were log transformed after adding a pseudocount. Gene expression for PERK signature genes was binned into normal samples or tumor samples and was plotted as a heat map using GENE-E. For each gene, a $p$ value and the difference in log transformed expression between cancer and normal was calculated.

For CREB3L1 expression across PERK and EMT bins, TCGA tumors were first binned into PERK-high or PERK-low bins based on if they were in the top 33 percentile or the bottom 33 percentile, respectively. For each PERK bin, EMT signaling was categorized as high or low if it was in the top 33 percentile of the bin or the bottom 33 percentile of the bin. For each tumor in each respective bin, CREB3L1 expression was plotted.

For primary tumor vs. metastasis comparisons, GSE20565, GSE20685, GSE7904, and GSE37444 were downloaded and quantile normalized. CREB3L1 expression across metastatic sites was compared to expression in primary tumors.

**Survival analysis**. Clinical survival analyses were performed through an online tool, KMplotter (kmplot.com)[47]. The information of data sets used was included in Fig. 1. To calculate a score for the signatures, average gene expression for each signature was determined. Patients were binned into those in the top tertile and those in the bottom tertile for the signature score. Data were plotted as a Kaplan–Meier curve.

**Statistics**. All statistical analyses were performed using Graphpad Prism software. All data unless otherwise specified are presented as mean ± s.e.m. A two-tailed Student's t-test was performed for comparisons between two groups of data. Significance of gene ontological enrichment was calculated using the hypergeometric test. Statistical significance of Kaplan–Meier survival analysis was calculated using the log-rank test. Significance of CREB3L1 binding to the CSPS genes was calculated using the binomial test. For each gene in tumor vs. normal comparisons, $p$ values were calculated using a Student's t-test, where only those that surpassed the Bonferroni-corrected threshold were considered significant.

**Data availability**. The data that support the findings of this study are available from the corresponding authors on reasonable request.

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

## Acknowledgements

We thank Dr David Pincus for the helpful discussions, and Dr Yuting Liu for the help on microarray data analyses. This research was supported by grants from the Richard and Susan Smith Family Foundation and the Breast Cancer Alliance (to P.B.G.), the Ludwig cancer research postdoctoral fellowship (to Y.-X.F.), and the National Science Foundation Graduate Research Fellowship Program (1122374; to E.S.S.).

## Author contributions

Y.-X.F., D.X.J. and P.B.G. designed the experiment. Y.-X.F., D.X.J. and E.S.S. conducted the experiments and analyzed the data. F.R. conducted the animal study. D.H.M. provided helpful discussions. Y.-X.F., D.X.J. and P.B.G. wrote the manuscript.

## Additional information

**Competing interests:** The authors declare no competing financial interests.

