## [Peer Review File · Nature Communications]

Reviewers' Comments:

Reviewer #1 (Remarks to the Author)

This manuscript describes the investigations into the PERK signaling pathway and cancer. The authors sought to identify downstream effectors of the PERK kinase that might be utilized to target the pathway without the unacceptable toxicities of targeting PERK itself. To do so, the authors identified a set of 23 genes in the pathway that were dysregulated between normal tissue and breast cancers. The authors selected CREB3L1 from this list since as a transcription factor it might play an important role in coordinating tumor related pathway expression. Subsequently, the authors demonstrated that manipulation of CREB3L1 expression altered extracellular matrix genes and FAK post-translational modification. In addition, they present data suggesting that CREB3L1 plays a role in invasion and migration. Finally, using chemical agents to prevent the processing of CREB3L1 to its active form, they demonstrate a potential role for metastatic disease, which appears to be associated specifically with the mesenchymal subtype of triple negative human breast cancer. The authors interpret these results to suggest that CREB3L1 is an important downstream modulator of the PERK pathway and a potential target for therapeutic intervention.

Concerns:

Second results section: The authors perform qPCR experiments of 6 of the proposed CREB3L1 targets. Why were these selected from the list of 14 CREB3L1 targets as defined by ChIP? Have the others been tested? Greater confidence in the interpretation of the results would be provided by presentation of the full panel of 14 gene, which could be included as a supplementary figure. Similarly, in the same section, growth curves supporting the statements made in the text should probably also be provided. Did the authors perform 3D culture of MDA-MBA-231 as well as SUM159? Performing these experiments in an independent cell line would provide greater confidence that the difference in invasion is a general property of CREB3L1, not a cell line specific affect. Finally, did the authors re-express the active form of CREB3L1 and show a rescue of the invasive phenotype?

In the "CREB3L1-induced ECM remodeling activates FAK" section, Fig. 3b should be Supplementary Fig. 3b. This section also seems to be unconnected in some ways from the rest of the manuscript. The authors demonstrate that the FAK phosphorylation defect in CREB3L1 knockdown cells can be rescued by addition of collagen 1. However, the connection between this and the interpretation that CREB3L1 mediates ECM remodeling is difficult to follow. Furthermore, the co-culture experiments do not provide convincing evidence that invasion, which is usually defined as penetration through basement membrane through proteolytic means, has been "rescued". Since one of the targets of CREB3L1 is MMP11, it is possible that the knockdown cells are deficient in proteolytic degradation of the ECM but are using tracts generated by the wildtype cells. Tumor cells can also traverse the basement membrane by amoeboid movement or as collective streams. So the data presented here is not conclusive and does not directly address the contention that ECM remodeling is a result of CREB3L1 expression.

Please provide p values for comparisons in all figures, including supplemental figures.

Figure 1c-e, it would be helpful to have the number of patients in each arm of the Kaplan-Meier plots listed to enable readers to better evaluate the data. What is the source of the data for KM plots? It does not appear to be listed in the manuscript text or material and methods section.

There are other breast cancer data sets available with clinical data. Similar results in other data sets would provide greater confidence in the results presented here.

Knock down of CREP3L1 was shown to reduce the expression of 6 of the 14 direct targets, as determined by ChIP-seq. What is the effect on the remaining 9 targets?

Reviewer #2 (Remarks to the Author)

The manuscript by Feng et al seeks to address mechanism whereby PERK might contribute to metastatic spread. Work from previous investigators have demonstrate that PERK regulates tumor progression and metastasis. Mechanisms include angiogenesis, SREBP signaling and redox control. Here the authors suggest CREB3L1 mediates metastatic spread. While the topic is of significance and could provide unique insights into this process, the work is rather preliminary and descriptive. It lacks the scientific rigor necessary to draw such conclusions. As noted below, there is a lack of evidence supporting a link between Perk and CREB3L1 and a lack of mechanistic insights into the potential regulation. The conclusions that general S1P inhibitors are useful and generate their effect through CREB3L1 are also insufficiently controlled to allow such conclusion. Additional specific suggestions are made below.

1. The authors conclude that a PERK specific signaling program is active in breast cancers. However, the evidence does not support this conclusion. The gene signature, many of which can be induced by PERK, are not PERK specific. Furthermore, there is no evidence that PERK is activated in any of the cancers. The author's should provide direct evidence for PERK activation in primary cancer and that genes are in fact PERK dependent; in short the gene signature is not sufficient to conclude that PERK is active and directly responsible for any phenotype described.
2. The authors assume CREB3L1 is downstream of PERK, yet no evidence is provided to support this conclusion. There is an assumption that it depends upon ATF4, but this is not interrogated. Overall, additional effort should be placed upon validated proposed mechanistic model, rather than assuming based upon correlative data.
3. Evidence that CREB3L1 regulates cancer metastasis is interesting, but there is a lack of mechanistic insight. Does this in fact reflect differential regulation of FAK?
4. With regard to CREB3L1 and metastasis, does its activation depend upon PERK activation?
5. Is CREB3L1 activation sufficient for the increase in circulating tumor cells and metastatic spread?
6. The experiments assessing CREB3L1 function by chemical inhibitors are of interest, but quite difficult to interpret given the large number of S1P/S2P substrates. CREB3L1 can rescue and this is an important control. However, to conclude specificity, additional rescue experiments are needed. For example, active ATF6 and SREBP1 are S1P/S2P substrates and should be examined for their capacity to rescue. The fact that SREBP1 is downstream of PERK, makes this an even more important experiment.
7. Given concerns regarding the role of PERK in activation CREB3L1, the authors should demonstrate that PERK is required for CREB3L1 activation and metastasis in their system. Use of small molecule inhibitors of PERK make this a very direct control and excellent way to connect specific points of interest in the proposed pathway.

We thank the reviewers for their helpful comments. We have addressed all of the reviewer concerns, resulting in a significantly strengthened manuscript. We provide below a point-by-point response to the reviewer's comments.

Reviewer #1 (Remarks to the Author):

Note: To help narrate our itemized response, we have numbered each of the reviewer's concerns below (the original comments were not numbered).

1. Second results section: The authors perform qPCR experiments of 6 of the proposed CREB3L1 targets. Why were these selected from the list of 14 CREB3L1 targets as defined by ChIP? Have the others been tested? Greater confidence in the interpretation of the results would be provided by presentation of the full panel of 14 gene, which could be included as a supplementary figure.

We have performed qPCR on the remaining seven CREB3L1 regulated genes (excluding CREB3L1 itself), and found that 10 out of 13 genes were significantly reduced in their expression in CREB3L1-knockdown cells. To evaluate if CREB3L1 regulates its own expression, we overexpressed a flag-tagged constitutively active form of the protein (CREB3L1^{Δ375-519}) in SUM159 cells. This experiment showed that the expression of endogenous full-length CREB3L1 was not changed.

These new experiments establish that 10 out of the 14 (~70%) putative CREB3L1 targets identified by ChIP are regulated by CREB3L1, but that CREB3L1 was not sufficient to upregulate its own expression. These new data are included in Fig.2c and Supplementary Fig. 2c of the revised manuscript.

2. Similarly, in the same section, growth curves supporting the statements made in the text should probably also be provided. Did the authors perform 3D culture of MDA-MBA-231 as well as SUM159? Performing these experiments in an independent cell line would provide greater confidence that the difference in invasion is a general property of CREB3L1, not a cell line specific affect. Finally, did the authors re-express the active form of CREB3L1 and show a rescue of the invasive phenotype?

We have provided the following new experiments to address the reviewer's concern:

- 1) We added the growth curves of SUM159 and MDA.MB.231 cells transduced with control or shRNAs targeting CREB3L1 in Supplementary Fig. 3c and 3d of the revised manuscript.**
- 2) We performed 3D culture of MDA.MB.231 cells transduced with control or shRNAs targeting CREB3L1 and found that knockdown of CREB3L1 significantly reduces the formation of invasive spheroids, as we had seen in SUM159 cells. These new data are included in Supplementary Fig. 3e of the revised manuscript.**
- 3) We re-expressed the active form of CREB3L1 (CREB3L1^{Δ375-519}) in CREB3L1-knockdown MDA.MB.231 cells and found that this rescues the reduced invasiveness caused by CREB3L1 knockdown. These new data are included in Supplementary Fig. 3h of the revised manuscript. Furthermore, we have shown in our initial submission that overexpression of CREB3L1^{Δ375-519} can rescue the decrease of cell invasion caused by AEBSF, an inhibitor of CREB3L1 activation (Fig. 4d of the revised manuscript). These experiments establish that re-expression of the active form of CREB3L1 rescues the invasive phenotype.**

3. In the "CREB3L1-induced ECM remodeling activates FAK" section, Fig. 3b should be Supplementary Fig. 3b. This section also seems to be unconnected in some ways from the rest of the manuscript. The authors demonstrate that the FAK phosphorylation defect in CREB3L1 knockdown cells can be rescued by addition of collagen 1. However, the connection between this and the interpretation that CREB3L1 mediates ECM remodeling is difficult to follow. Furthermore, the co-culture experiments do not provide convincing evidence that invasion, which is usually defined as penetration through basement membrane through proteolytic means, has been "rescued". Since one of the targets of CREB3L1 is MMP11, it is possible that the knockdown cells are deficient in proteolytic degradation of the ECM but are using tracts generated by the wildtype cells. Tumor cells can also traverse the basement membrane by

amoeboid movement or as collective streams. So the data presented here is not conclusive and does not directly address the contention that ECM remodeling is a result of CREB3L1 expression.

We thank the reviewer for their comments about this section. In response, we have made the following revisions in the manuscript to further clarify the mechanism by which CREB3L1 activates FAK to promote invasion:

- 1) **We changed title of this section to “CREB3L1-induced ECM deposition activates FAK” in the revised manuscript.**
- 2) **We report in the revised manuscript that CREB3L1 induces the expression of ECM genes, including COL1A2 and FN1, and activates FAK. Functionally, we show that addition of type I collagen (COL1A1 and COL1A2) concomitantly rescued the decrease of FAK phosphorylation (Supplementary Fig. 4c of the revised manuscript) and cell migration (Fig. 2i of the revised manuscript) in CREB3L1-inhibited cells. We also show that regulation of cell motility plays a key role in CREB3L1-driven invasion, since knockdown of CREB3L1 also markedly inhibits cell migration (Supplementary Fig. 4b of the revised manuscript). As the reviewer mentioned, it was also possible that CREB3L1 promotes ECM degradation through MMP11. But based on these new findings, we concluded that CREB3L1 promotes cancer cell migration and invasion primarily through ECM deposition-triggered FAK activation.**
- 3) **We replaced the co-culture experiment with two experiments that directly address the role of ECM remodeling in promoting invasion, and provide some mechanistic insight into how CREB3L1 promotes invasion. In the first experiment, we show that CREB3L1-knockdown cells have reduced motility (Supplementary Fig. 4b of the revised manuscript). In the second experiment, we supplemented CREB3L1-knockdown cells with type I collagen or FN1 and found they rescue cell migration (Fig. 2i of the revised manuscript). Furthermore, in our initial submission, we had shown that CREB3L1 knockdown cells have reduced FAK-activity and that this reduced activity can be rescued by supplementing cells with type I collagen or FN1 (Supplementary Fig. 4c of the revised manuscript). Taken together, these data indicate that CREB3L1 promotes invasion by depositing new matrix to activate FAK, resulting in increased cell motility.**

4. Please provide p values for comparisons in all figures, including supplemental figures.

We have provided p values wherever needed.

5. Figure 1c-e, it would be helpful to have the number of patients in each arm of the Kaplan-Meier plots listed to enable readers to better evaluate the data. What is the source of the data for KM plots? It does not appear to be listed in the manuscript text or material and methods section. There are other breast cancer data sets available with clinical data. Similar results in other data sets would provide greater confidence in the results presented here.

All data of the Kaplan-Meier (KM) plots were derived from an online tool, Kaplan-Meier Plotter (KMplot.com). We have added sample size information to Fig. 1c-e of the revised manuscript, and provided the information of data sets in the Method section.

The data of each KM plot in Fig. 1 of the revised manuscript was derived from multiple data sets. Taking Fig.1c (breast cancer) as an example, it is in fact a meta-analysis on data sets including GSE17907 (n=38), GSE9195 (n=77), GSE20685 (n=327), GSE16446 (n=107), and GSE19615 (n=105). Therefore, the results presented in Fig.1c-e are general observations that validate across multiple independent data sets.

6. Knock down of CREP3L1 was shown to reduce the expression of 6 of the 14 direct targets, as determined by ChIP-seq. What is the effect on the remaining 9 targets?

We have addressed this issue in our response to the reviewer’s first comment above.

Reviewer #2 (Remarks to the Author):

1. The authors conclude that a PERK specific signaling program is active in breast cancers. However, the evidence does not support this conclusion. The gene signature, many of which can be induced by PERK, are not PERK specific. Furthermore, there is no evidence that PERK is activated in any of the cancers. The author's should provide direct evidence for PERK activation in primary cancer and that genes are in fact PERK dependent; in short the gene signature is not sufficient to conclude that PERK is active and directly responsible for any phenotype described.

To address the reviewer's first point, we never intended to suggest that a "PERK-specific" signaling program is active in breast cancers. The reviewer appears to have transmuted the language used in our manuscript ("cancer-specific PERK signaling") to "PERK-specific cancer signaling". Our goal was to identify a cancer-specific PERK signaling signature (PERK pathway genes up-regulated in cancer compared to normal tissue), which we in fact do in Figure 1 of the paper. In no place in our manuscript do we claim that the genes we identified (CSPS, fig. 1a,b of the revised manuscript) were specifically regulated by PERK. In fact, as is shown in Fig. 5 of the revised manuscript, CREB3L1 is regulated by both ATF4 (downstream of PERK signaling) and Fra-1 (downstream of EMT signaling).

With respect to the reviewer's second point, it would be inappropriate for our study to assess the already established functional importance of PERK in cancer. Activation of PERK in cancers has been reported by a great number of previous publications, and was the motivation for multiple pharmaceutical companies to invest continuous efforts to generate PERK inhibitors (Axten et al, 2012. *J. Med. Chem*; Atkins et al, 2013. *Cancer Research*). In addition, our previous paper has already shown that PERK and its signaling pathway components are activated in primary breast cancers (Feng et al, 2014. *Cancer Discovery*). A similar finding was reported by another paper (Chen et al, 2014. *Nature*).

To further address the reviewer's concerns, we provided the following new experiments in the revised manuscript: (i) we performed immunohistochemistry (IHC) staining on primary breast cancers using a phospho-PERK specific antibody. We show in the revised manuscript that PERK was phosphorylated—therefore activated—in over 70% of the samples tested (Supplementary Fig. 1a of the revised manuscript); (ii) we used a small-molecule inhibitor of PERK activity, GSK2656157, and found ~75% of CSPS genes were downregulated following treatment (Supplementary Fig. 1b of the revised manuscript).

2. The authors assume CREB3L1 is downstream of PERK, yet no evidence is provided to support this conclusion. There is an assumption that it depends upon ATF4, but this is not interrogated. Overall, additional effort should be placed upon validated proposed mechanistic model, rather than assuming based upon correlative data.

The reviewer appears to have overlooked the data already included in Fig. 5 of our original submission. In Fig. 5, we showed that ATF4 knockdown significantly reduces the expression of CREB3L1 in two independent breast cancer cell lines. Since ATF4 is a well-established factor downstream of PERK, these results provide direct evidence that PERK functionally regulates the expression of CREB3L1.

In the revised manuscript, we also included additional findings that provide further evidence addressing this point. Specifically, we now show that CREB3L1 expression is significantly inhibited when cells are treated with a PERK inhibitor (Supplementary Fig. 1b and 8 of the revised manuscript) – a result again consistent with a model placing CREB3L1 downstream of PERK.

3. Evidence that CREB3L1 regulates cancer metastasis is interesting, but there is a lack of mechanistic insight. Does this in fact reflect differential regulation of FAK?

We show in the revised manuscript that CREB3L1 promotes FAK activation through phosphorylation on Tyr397 and Tyr576/577 (Fig. 2h of the revised manuscript). While we would

have liked to directly measure FAK phosphorylation in lung metastasis formed by cancer cells, we found this to be technically unfeasible because the lung metastases formed by CREB3L1-KD cancer cells were essentially non-existent; this made it impossible to compare the levels of FAK phosphorylation between control and CREB3L1-KD metastases *in vivo*.

At a mechanistic level, we show that CREB3L1's activation of FAK is mediated through ECM production. Moreover, we show in the revised manuscript that addition of type I collagen or fibronectin is sufficient to rescue the decrease in FAK activation and invasion observed upon inhibition of CREBL1 in SUM159 cancer cells (Fig. 2i and Supplementary Fig.4c of the revised manuscript).

4. With regard to CREB3L1 and metastasis, does its activation depend upon PERK activation?

To address the reviewer's question, we inhibited PERK with a small-molecule inhibitor, GSK2656157, and performed western blot to assay the expression level of both full-length (uncleaved) and the N-terminus (cleaved) of CREB3L1. We found that both uncleaved and cleaved CREB3L1 is decreased in PERK-inhibited cells (Supplementary Fig. 8 of the revised manuscript), and that the ratio of uncleaved CREB3L1 to cleaved CREB3L1 is unchanged. These results indicated that PERK induces the expression of CREB3L1 (through ATF4), but does not affect the activation of CREB3L1.

5. Is CREB3L1 activation sufficient for the increase in circulating tumor cells and metastatic spread?

To address the reviewer's question, we show in Fig. 4e of the revised manuscript that CREB3L1 activation is not sufficient to increase metastatic spread, since overexpression of a constitutively active form of CREB3L1 (CREB3L1^{Δ375-519}) in MDA.MB.231-LM2 cells cannot increase lung metastasis of this line. Consistently, the circulating tumor cell count was not changed in mice implanted with cancer cells overexpressing CREB3L1^{Δ375-519} when compared to those with control cancer cells (data not shown). However, over-expression of CREB3L1^{Δ375-519} can rescue the decrease of FAK activation (Fig. 4c of the revised manuscript), cell invasion (Fig. 4d of the revised manuscript), and metastatic spread (Fig. 4e of the revised manuscript) in AEBSF-treated cells. These data suggest that CREB3L1 activation is required but not sufficient for the increase in circulating tumor cells and metastatic spread of invasive cancer cells.

6. The experiments assessing CREB3L1 function by chemical inhibitors are of interest, but quite difficult to interpret given the large number of S1P/S2P substrates. CREB3L1 can rescue and this is an important control. However, to conclude specificity, additional rescue experiments are needed. For example, active ATF6 and SREBP1 are S1P/S2P substrates and should be examined for their capacity to rescue. The fact that SREBP1 is downstream of PERK, makes this an even more important experiment.

We thank the reviewer for recommending these controls. In our original submission, we have attempted to address the concerns of chemical specificity in the discussion section. To more thoroughly address these concerns, we added the recommended experiments in the revised manuscript. We found that overexpression of the constitutively active form of ATF6 or SREBP1 is not sufficient to rescue the reduction in invasion following treatment with AEBSF (Supplementary Fig. 6e and 6f of the revised manuscript). This is in contrast to what we observed in cells overexpressing constitutively active CREB3L1, which was sufficient to rescue the reduction in invasion (Fig. 4d of the revised manuscript). Collectively, these observations suggest that S1P inhibitors, such as AEBSF and PF429242, suppress invasion and metastasis predominantly by blocking activation of CREB3L1.

7. Given concerns regarding the role of PERK in activation CREB3L1, the authors should demonstrate that PERK is required for CREB3L1 activation and metastasis in their system. Use of small molecule inhibitors of PERK make this a very direct control and excellent way to connect specific points of interest in the proposed pathway.

To address the reviewer's concerns, we and others have previously shown that activation of the PERK pathway is required for metastasis (Feng et al., 2014. *Cancer Discovery*; Dey et al., 2015. *JCI*). In our study, we have shown that a small-molecule PERK inhibitor effectively inhibits the lung metastatic spreading of cancer cells. These can address the concern that "the authors should demonstrate that PERK is required for ... metastasis in their system."

We thank the reviewer for suggesting using PERK inhibitors to validate the role of PERK in regulating CREB3L1. In Supplementary Fig. 1a and 8 of the revised manuscript, we show that inhibition of PERK via a small-molecule inhibitor can reduce the expression of CREB3L1 but does not affect the activation of CREB3L1. Combined with the data that ATF4 is required for the expression of CREB3L1 (Fig. 5a of the revised manuscript), we conclude that the PERK-ATF4 pathway is required for transcriptional activation of CREB3L1.

Reviewers' Comments:

Reviewer #1:

Remarks to the Author:

This manuscript is a revised version of a study to investigate the potential of the PERK downstream molecule CREB3L1 as a target for clinical intervention in cancer. The authors have responded to a number, but not all, of the previous criticisms. However, significant concerns still exist regarding the the data presentation and validity of the analytic interpretation:

Lines 80-82 in results section: The authors state that "expression of these genes depended on PERK activity, since inhibition of PERK...led to a significant decrease in their expression." This statement is incorrect and inaccurate based on the figure presented. 4 of the genes are presented as having non-significant changes. SLC2A6 is shown as upregulated. This statement needs to be edited to accurately reflect the data.

Line 88: the data from supplemental figure 1b supports 19 of the genes being PERK dependent, not 23.

Line 120: Please provide statistical analysis of the results for supplemental figure 3g&h.

Lines 138-145: The authors have convincingly demonstrated that addition of collagen can increase FAK phosphorylation and rescue the migratory phenotype. However, they have not directly and formally shown that it is the expression of collagen from the tumor cell that is responsible. That would require specific ectopic expression of these genes in the CREB3L1 knockdown cells to prove that there is not some other mechanism that exogenous ECM activates. The authors need either to soften the interpretation of their data or perform these experiments to convincingly prove this hypothesis.

Line 153: Please provide statistical results of the tumor weight analysis. Also, based on the variance seen within the control group, is five animals per group sufficient to provide enough statistical power to address the question regarding the effect of CREB3L1 on tumor growth?

Figure 3: Panels A&B, have these experiments been replicated? Or are they single experiments with only 5 animals per group? Panel C, the Y-axis is labeled as "GFP+ cancer cells/Field". How many fields per animal were scored? What percentage of the lung cross section was analyzed? Are the investigators identifying predominantly single disseminated cells or multicellular metastases? Dissemination of tumor cells and formation of clinically relevant metastases are different and the term "metastasis" should not be used for cells that have not completed the metastatic process. The results of panels b & C do not make sense if in the control the average is one cell per field. The amount of signal seen in figure panel B looks like widespread, dispersed cells, such as seen immediately after tail vein injection, rather than discrete multicellular mets as shown in panel C. In addition, N=10 in panel C. Why are the additional 5 animals not included in panel B?

Supplemental figure 6: Please provide statistical analyses on all of the various comparisons for the histogram plots. It is important to know what is not significant as well as what is significant.

Figure 4e: was histology performed on the lungs to count multicellular metastatic lesions? AEBSF treated delta(375-519) animals appear to have a reduction lung radiance, which may actually indicate a suppression of metastatic disease. Due to the logarithmic scale and inherent variability in bioluminescence assays it is difficult to interpret this result definitively using only the luciferase assay.

Reviewer #2:

Remarks to the Author:

Most concerns addressed adequately.

We thank the reviewer for their helpful comments. As detailed below, we have addressed each of the concerns by either revising the text or providing additional analyses where requested by the reviewer – resulting in a significantly improved revised manuscript. Please find below our point-by-point response to each of the reviewer’s comments.

Lines 80-82 in results section: The authors state that “expression of these genes depended on PERK activity, since inhibition of PERK...led to a significant decrease in their expression.” This statement is incorrect and inaccurate based on the figure presented. 4 of the genes are presented as having non-significant changes. SLC2A6 is shown as upregulated. This statement needs to be edited to accurately reflect the data.

We thank the reviewer for this observation and have revised the text to precisely state that 18 of the 23 CSPA genes are positively regulated by PERK.

Line 88: the data from supplemental figure 1b supports 19 of the genes being PERK dependent, not 23.

The reviewer is correct in indicating that 19 of the CSPA genes are regulated by PERK, of which 18 are positively regulated and 1 is negatively regulated by PERK. We have revised the text to correct the misstatement.

Line 120: Please provide statistical analysis of the results for supplemental figure 3g&h.

We have revised the manuscript to provide statistical analyses of the results in Fig. S3e and S3h. [Note: Since Fig. S3g is a western blot, we have assumed that the reviewer intended to specify Fig. S3e and Fig. S3h in their remark.]

Lines 138-145: The authors have convincingly demonstrated that addition of collagen can increase FAK phosphorylation and rescue the migratory phenotype. However, they have not directly and formally shown that it is the expression of collagen from the tumor cell that is responsible. That would require specific ectopic expression of these genes in the CREB3L1 knockdown cells to prove that there is not some other mechanism that exogenous ECM activates. The authors need either to soften the interpretation of their data or perform these experiments to convincingly prove this hypothesis.

We have revised the text to emphasize that the ECM was exogenously applied to rescue the decrease in cell migration. Also, as suggested by the reviewer, we have revised the text to soften our interpretation of this result.

Line 153: Please provide statistical results of the tumor weight analysis. Also, based on the variance seen within the control group, is five animals per group sufficient to provide enough statistical power to address the question regarding the effect of CREB3L1 on tumor growth?

The reviewer raises a valid point, which we have addressed by providing a statistical analysis of the results in Fig. S5. These analyses indicate that there is no statistically significant difference in tumor weight between the control and CREB3L1-KD groups. In addition, given the variance seen within the control group, the statistical power provided by 5 animals per group is 0.98 if at least a 2-fold change is expected between control and CREB3L1-KD groups (even if the difference between groups is only 1.5-fold, the power is still over 0.8). Considering the fact that there is an over 400-fold difference in lung metastasis between control and CREB3L1-KD mice, this power analysis indicates that it is reasonable to conclude that CREB3L1 primarily affects metastasis but not tumor growth.

Figure 3: Panels A&B, have these experiments been replicated? Or are they single experiments with only 5 animals per group? Panel C, the Y-axis is labeled as “GFP+ cancer cells/Field”. How many fields per animal were scored? What percentage of the lung cross section was analyzed? Are the investigators identifying predominantly single disseminated cells or multicellular metastases? Dissemination of tumor cells and formation of clinically relevant metastases are different and the term “metastasis” should not be used for cells that have not completed the metastatic process. The results of panels b & C do not make sense if in the control the average is one cell per field. The amount of signal seen in figure panel B looks like widespread, dispersed cells, such as seen immediately after tail vein injection, rather than discrete multicellular mets as shown in

panel C. In addition, N=10 in panel C. Why are the additional 5 animals not included in panel B?

- 1) We have performed each of the experiments in Fig. 3a and 3b at least two times with 5 animals per group for each experiment. Fig. 3a and 3b show representative results from one experiment.
- 2) We apologize for the confusion caused by our imprecise labeling of the Y-axis, and have revised the axis label to “Relative lung metastasis”, more precisely reflecting the quantification which was described in the Methods section of the original submission. In brief, we used ImageJ to quantify the GFP+ area in each image, and normalized the results of each image to the mean of the control group. We did not count the number of GFP+ cells.
- 3) Two fields per animal were scored, and a total of 10 fields were scored per group (that is why N=10 in panel C). On each lung cross-section, ~5% of the area was scored. It is necessary to clarify that it is much more reliable to quantify the overall lung metastases using luminescence measurements, since the whole lung was assayed and the potential issues caused by sampling is completely avoided. The primary goal of GFP staining is to qualitatively confirm the existence of cancer cells in the lung.
- 4) In the control group, the majority of GFP+ area (>80%) is multicellular, therefore falling into the category of “clinically relevant metastases”.
- 5) It can be seen in a great number of papers that the luminescence signal observed in a mouse lung collected from a late-stage spontaneous metastasis assay (what we did in Fig. 3b) is usually widespread, even though the cancer cells metastasized to the lung are isolated and discrete. If there are a sufficient number of metastases dispersed throughout the lungs (as is the case for our experiments), images of luminescence and IHC staining cannot be compared because of differences in the resolution of the respective experimental techniques.

Supplemental figure 6: Please provide statistical analyses on all of the various comparisons for the histogram plots. It is important to know what is not significant as well as what is significant.

We have added additional comparisons in Fig. S6 of the revised manuscript, and they are listed below. However, we would note that only a subset of the comparisons below is relevant to our scientific hypothesis.

S6B Ctrl vs Ctrl+OE = NS, p=0.0853

S6C Ctrl vs OE = NS, p=0.1536

S6C Ctrl vs OE+AEBSF is significant, p=0.008

S6C OE vs OE+AEBSF = NS, p=0.0853

S6D – Ctrl vs OE is NS, p=0.8328

S6D – Ctrl vs Ctrl+PF is significant, p=0.0029, but there is only a 30% reduction in growth, whereas there is a >95% reduction in invasion

S6D – Ctrl vs OE+PF is significant, p=0.0001, but while there is a 20% reduction in invasion, most of it can be accounted for by the 30% reduction in growth

S6D – OE vs OE+PF is NS, p=0.6743

S6F – Ctrl untreated vs SREBP1OE untreated is NS, p=0.3683

S6F – Ctrl untreated vs ATF6 untreated is NS, p=0.3845

S6F – Ctrl+AEBSF vs SREBP1OE+AEBSF is NS, p=0.235

S6F – Ctrl+AEBSF vs ATF6OE+AEBSF is NS, p=0.1981

(NS: not significant)

Figure 4e: was histology performed on the lungs to count multicellular metastatic lesions? AEBSF treated delta(375-519) animals appear to have a reduction lung radiance, which may actually indicate a suppression of metastatic disease. Due to the logarithmic scale and inherent variability in bioluminescence assays it is difficult to interpret this result definitively using only the luciferase assay.

We did not perform IHC analysis for Fig. 4e because it would only serve to confirm the existence of cancer cells in lung -- which was already shown to be the case in Fig. 3c. Luminescence

imaging is a far more accurate method to quantify lung metastasis, and we have provided this method of quantification in Fig. 4e. We note that there was only a 1.7-fold difference ($p=0.43$) between the geometric means of control-treated delta(375-519) and AEBSF-treated delta(375-519) animals. In contrast, there was a >14-fold difference ($p<0.05$) between the geometric means of vehicle-treated control and AEBSF-treated control animals. These observations indicate that overexpression of delta(375-519) effectively rescues the decreased metastasis caused by treatment with AEBSF.

Reviewers' Comments:

Reviewer #1:

Remarks to the Author:

The authors have responded positively to my concerns. I believe that this manuscript is for the most part ready for publication. I would suggest that the details of some of the experiments, that were provided in response to my concerns, be added to the materials and methods section for clarity.

The one remaining concern I have is regarding the use only luciferase signal as a measure of macrometastatic disease. The authors should be aware that luciferase signal is not, in our experience, a good measure of macroscopic metastatic disease. Hypoxic or necrotic metastases are not particularly bioluminescent. Thus reliance on only bioluminescence signal can underestimate the metastatic burden. As a result, in my opinion, confirmation by enumeration of metastases, by either surface count or histological analysis, provides significantly more confidence in the result. However, if the editor believes that the authors have provided sufficient evidence to make this analysis unnecessary I have no additional concerns.

The authors have responded positively to my concerns. I believe that this manuscript is for the most part ready for publication. I would suggest that the details of some of the experiments, that were provided in response to my concerns, be added to the materials and methods section for clarity.

We have provided additional details of the experiments into the revised manuscript.

The one remaining concern I have is regarding the use only luciferase signal as a measure of macrometastatic disease. The authors should be aware that luciferase signal is not, in our experience, a good measure of macroscopic metastatic disease. Hypoxic or necrotic metastases are not particularly bioluminescent. Thus reliance on only bioluminescence signal can underestimate the metastatic burden. As a result, in my opinion, confirmation by enumeration of metastases, by either surface count or histological analysis, provides significantly more confidence in the result. However, if the editor believes that the authors have provided sufficient evidence to make this analysis unnecessary I have no additional concerns.

Bioluminescence is a well-acknowledged approach to quantify metastasis using the LM2 model that we used in our paper (Minn AJ, et al., Nature 2005; Gupta GP, et al., Nature 2007). In addition, it is known that signal of bioluminescence is well-correlated with the number of macrometastases (Klerk CP, et al., Biotechniques 2007).

As indicated in our second round of review, we did in fact apply sectioning/IHC as an additional method to quantify metastasis (Figure 3C). This alternative method of quantification confirmed the results that were collected using bioluminescence imaging (Figure 3B).